# CO-PILOT: COllaborative Planning and reInforcement Learning On sub-Task curriculum

**Shuang Ao[1], Tianyi Zhou[2,3], Guodong Long[1], Qinghua Lu[4], Liming Zhu[4], Jing Jiang[1]**
University of Technology Sydney[1];
University of Washington, Seattle[2]; University of Maryland, College Park[3];
CSIRO's Data61, Australia[4]
`shuang.ao@student.uts.edu.au, tianyizh@uw.edu,`
`{guodong.long, jing.jiang}@uts.edu.au,`
`{qinghua.lu, liming.zhu}@data61.csiro.au`

## Abstract

Goal-conditioned reinforcement learning (RL) usually suffers from sparse reward and inefficient exploration in long-horizon tasks. Planning can find the shortest path to a distant goal that provides dense reward/guidance but is inaccurate without a precise environment model. We show that RL and planning can collaboratively learn from each other to overcome their own drawbacks. In "**CO-PILOT**", a learnable path-planner and an RL agent produce dense feedback to train each other on a curriculum of tree-structured sub-tasks. Firstly, the planner recursively decomposes a long-horizon task to a tree of sub-tasks in a top-down manner, whose layers construct coarse-to-fine sub-task sequences as plans to complete the original task. The planning policy is trained to minimize the RL agent's cost of completing the sequence in each layer from top to bottom layers, which gradually increases the sub-tasks and thus forms an easy-to-hard curriculum for the planner. Next, a bottom-up traversal of the tree trains the RL agent from easier sub-tasks with denser rewards on bottom layers to harder ones on top layers and collects its cost on each sub-task train the planner in the next episode. CO-PILOT repeats this mutual training for multiple episodes before switching to a new task, so the RL agent and planner are fully optimized to facilitate each other's training. We compare CO-PILOT with RL (SAC, HER, PPO), planning (RRT*, NEXT, SGT), and their combination (SoRB) on navigation and continuous control tasks. CO-PILOT significantly improves the success rate and sample efficiency. Our code is available at https://github.com/Shuang-AO/CO-PILOT.

## 1 Introduction

Although AI can surpass humans on certain tasks, humans still perform much better in making sequential decisions via learning from interactions with the environment. Reinforcement learning (RL) [50] aims to bridge this gap by learning to optimize the trajectories of agents (e.g., controllers, robots, game players, self-driving cars, etc) to achieve the maximal return. However, in complicated long-horizon tasks, RL usually suffers from poor sample efficiency and costly data collection. Moreover, the data quality is often low due to sparse rewards when rollouts fail and cannot provide informative feedback. Model-based RL and off-policy RL improve the sample complexity with the price of extra biases, causing unstable and brittle optimization. Instead of reaching a single goal, goal-conditioned RL [27] learns one model for any given goal input to its model(s). However, it needs to be trained to reach many possible goals, and the resulted model's performance still degrades drastically for distant goals.

35th Conference on Neural Information Processing Systems (NeurIPS 2021).

Instead, planning algorithms are usually more robust and effective on long-horizon tasks. Given a distance metric, they discretize the state space to a grid/graph and seek for the shortest collision-free path between states using graph search such as Dijkstra's algorithm or A* [22]. Thereby, it only needs a local policy to navigate between consecutive states on the path. However, it is challenging to learn or estimate the distance accurately in complicated tasks such as mazes. Moreover, planning every step on the path is as difficult as the original RL and requires fine-grained discretization impractical for high-dimensional states. Planning only a few milestone states leaves the RL agent to solve relatively long-horizon sub-tasks. Although sampling-based search heuristics can build a graph with a better exploration-exploitation trade-off, they are not optimized for the RL policy. [13] adapt planning to a learned RL policy, which can provide distances estimated from its replay buffer, but the performance largely depends on the RL policy and its exploration.

A critical insight of this paper is that planning at even a coarse level can be used for reward shaping and substantially improves RL on long-horizon tasks with sparse reward. In contrast, experiences of the RL agent on the planned sub-tasks can improve the distance metric of planning to produce better paths/sub-tasking for the RL agent. Hence, the RL agent and path-planner can provide dense and informative feedback to train each other. Thus, combining their strengths helps to overcome the bottleneck of each one and improve their exploration efficiency.

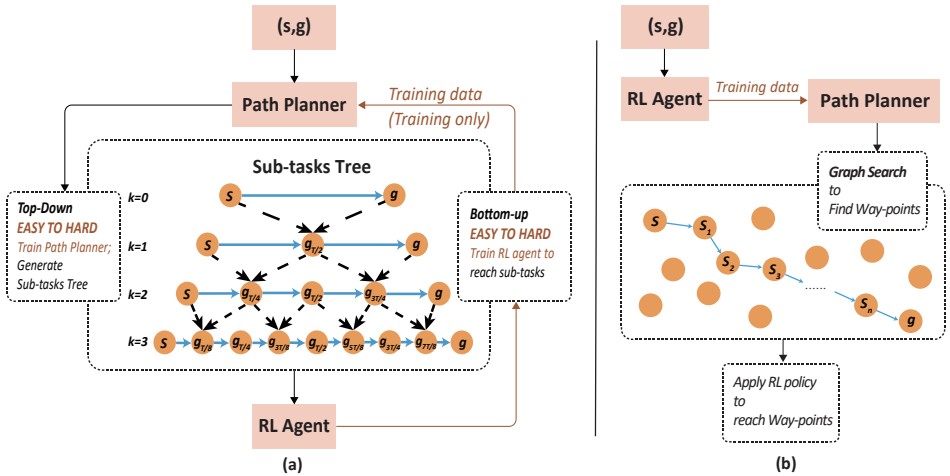

Figure 1: **(a)** Mutual training between RL and planner in CO-PILOT. The planner is trained to recursively decompose a task $(s, g)$ to a sub-task tree of coarse-to-fine min-cost sequences of sub-tasks. While this top-down construction forms an easy-to-hard curriculum to train the planner, a bottom-up traversal of those sub-tasks forms an easy-to-hard curriculum for RL. The planned sub-tasks provides dense rewards enabling more efficient RL, while RL's cost on each sub-task is used to train the planner for producing more cost-efficient sub-tasks for RL. For comparison, **(b)** describes how SoRB [13] combines RL with planning, which does not adopt such mutual training scheme and the sub-task curricula. In both diagrams, the brown arrows only happen in the training phase.

In this paper, we propose "**CO-PILOT**", a collaborative learning scheme between planning and goal-conditioned RL. As illustrated in Figure 1 (a), it trains each model under the other's guidance along a curriculum of sub-tasks. Unlike most existing planning methods, we train a planning policy to recursively decomposes a task into two easier sub-tasks, which finally yields a tree containing coarse-to-fine trajectories of sub-goals to the final goal. The tree naturally forms a curriculum for more effective training. During the top-down tree construction, we start from training the planner to find the shortest path on a coarser graph with fewer sub-goals, which is an easier training task, and gradually request it to generate detailed paths with denser sub-goals. We measure the distance by the cost of an RL agent navigating between consecutive sub-goals, so the planner is optimized to produce the most efficient path for the RL agent.

With the sub-goal tree constructed, we then train the goal-conditioned RL agent by a bottom-up curriculum, starting from easier sub-tasks with dense reward along the path and gradually enforcing the RL agent to navigate between more distant sub-goals. The sub-goals previously generated by the planner now provide an accurate reward shaping since they constitute cost-efficient paths for the RL agent. As a byproduct of rollouts on the sub-tasks, the RL policy helps to explore the environment topology and collect cost data between states to refine the distance metric for planning. Hence, the

top-down (bottom-up) curriculum training of planner (RL agent) eases the training on the original tasks and collects more informative feedback to train the RL agent (planner). CO-PILOT repeats the above procedures for episodes of mutual boosting between the two until they are fully optimized for the other. In experiments, we apply CO-PILOT to navigation and continuous control. Compared to existing RL, planning and combining them, CO-PILOT significantly improves the sample efficiency and the final success rate for long-horizon tasks.

## 2 Related Work

**RL:** Goal-conditioned RL [40, 27, 45] takes a goal as an additional input to its model(s) and aims to handle different goals/tasks using the same policy. However, it requires more exploration and expensive training on various possible goals, and it still easily fails to reach distant goals in practice. Goal-relabeling and reward shaping [3, 40, 19] have been commonly studied to mitigate these issues. Recent methods [16, 55] improve it by learning a compact representation of the goal space. The goal-conditioned value function $V(s|g)$ naturally provides an ideal distance metric for shortest path planning. [10, 54] propose to train RL policy on a curriculum of environments adaptive to the RL. In CO-PILOT, we train a sub-goal tree planner to generate a sample-efficient and adaptive training curriculum that trains goal-conditioned to reach distant goals progressively. On the other hand, the goal-conditioned policy's cost on the planned paths is used to improve the planning policy.

**Planning:** Planning [50, 34] are more effective in addressing long-horizon tasks in practice [37, 29]. It usually refers to dynamic programming that finds the optimal path between two nodes on a graph. Planning methods in RL, e.g., value/policy iteration [33, 37], utilize or learn an environment model to improve the RL policy. Compared to a reactive policy, an advantage of planning is that the planned trajectory provides a global view of future steps. However, learning the environment model usually requires expensive exploration of state space [12, 35, 24, 39], structured and compact modeling of the environment/graph [7, 41], and an accurate distance metric [13]. CO-PILOT overcomes these limitations by "**learning to plan**", which trains a planning policy to produce sequences of sub-tasks that minimize the cost of the RL agent for completing the sequences. Specifically, we adopt recent sub-goal tree (SGT) planning [25] to generate tree-structured coarse-to-fine sequences of sub-tasks, instead of searching on a pre-built hierarchical partition tree of the state space as many hierarchical planning methods [39]. Moreover, we use the RL agent's time cost on the sub-tasks as the distance metric to train the planner, which is more accurate and adaptive. Furthermore, every-step planning is not necessary since a few sub-tasks may already suffice to provide dense rewards and guidance to efficiently train the RL agent, so the planning in CO-PILOT can be much easier. Compared to SGT and other hierarchical planning methods, the collaborative training between RL and "learning to plan" along easy-to-hard curricula in CO-PILOT can efficiently improve the performance of both RL and planning without heavily relying on prior knowledge.

**Combine RL with Planning:** A line of recent works [1, 36, 49, 46] embeds a planning model as one part of an RL agent's model and train it together with the RL policy in an end-to-end manner. [9, 15, 44] find that combining the two can help agents to reach distant goals in specific tasks. [13, 44] propose planning strategies with graph search based on the replay buffer of experiences from a given RL policy. [46] proposes to use Monte-Carlo tree search when planning in latent space to achieve a better optimization on value function. These results inspire our work, but our primary difference is the mutual training between RL and planning, which does not require either a pre-trained policy or strong heuristics about the distance metric. In CO-PILOT, both are trained from scratch and can mutually boost and guide each other's training via an auto-generated curriculum of easy-to-hard sub-tasks. This mutual training leads to a principled learning framework adaptive to a vast amount of potential applications.

**Hierarchical RL:** Hierarchical RL (HRL) [26, 18, 38, 32] learns a sequence of primeval policies, e.g., policies for low-level skills or easier sub-tasks, and then sequentially composes them to form a high-level policy addressing complicated tasks. Another kind of HRL methods transfers the knowledge of a morphologically simpler agent to a more complex one [23, 8]. HRL for goal-reaching tasks has been studied in [13, 38]. A key challenge in HRL is how to define low-level skills or sub-goals [11, 17, 48, 2]. In CO-PILOT, we train a planning policy to automatically propose sub-tasks of increasing difficulty to train the RL agent. Therefore, RL agent starts from learning how to reach nearby sub-goals and progressively improve its skills for completing long-horizon tasks.

# 3 CO-PILOT

## 3.1 Goal-conditioned Reinforcement Learning

Goal-conditioned RL or multi-goal RL learns a policy that can be adapted to different goals. Given the state space $\mathcal{S}$, the action space $\mathcal{A}$, and the goal space $\mathcal{G}$, a goal-conditioned policy is a mapping $\pi(a|s, g) : \mathcal{S} \times \mathcal{G} \mapsto \mathcal{A}$ that outputs an action $a$ (or probabilities $\Pr(a|s, g)$ over actions $a \in \mathcal{A}$) given a state-goal pair $(s, g)$. An RL agent uses $\pi(a|s, g)$ to interact with an environment described by a Markov decision process (MDP) $\{\mathcal{S}, \mathcal{A}, \mathcal{G}, p, r, \gamma\}$, where $p(s'|s, a) \triangleq \Pr(s_{t+1} = s'|s_t = s, a_t = a)$ is the transition probability for the agent from state $s$ to $s'$ after taking action $a$, $r(s, a|g) : \mathcal{S} \times \mathcal{A} \times \mathcal{G} \mapsto \mathbb{R}$ is a reward function, and $\gamma \in [0, 1]$ is a discount factor.

In each episode, the agent starts from an initial state $s_0 \sim p_0(s)$ and aims to reach a given goal $g \in \mathcal{G}$. In every time step $t$, it takes an action $a_t = \pi(a|s_t, g)$ (deterministic) or $a_t \sim \pi(a|s_t, g)$ (stochastic), receives a reward $r(s_t, a_t|g)$, and moves to a new state $s_{t+1} \sim p(s'|s_t, a_t)$. RL aims to learn a policy $\pi$ maximizing the expected return $\mathbb{E}_{(s_0, g)}[\mathbb{E}_\pi(R_0)]$. Define the action-value function $Q(s, a|g) \triangleq \mathbb{E}(R_t|s_t = s, a_t = a, g)$, the optimal policy $\pi^*$ achieves the maximal $Q(s, a|g)$ for any feasible $(s, a, g)$. Define the value function $V(s|g) \triangleq \mathbb{E}(R_t|s_t = s, g) = \mathbb{E}_{a \sim \pi}[Q(s, a|g)] = \sum_{a \in \mathcal{A}} \pi(a|s, g)Q(s, a|g)$. Directly maximizing the expected return or $V$ w.r.t. $\pi$ results in the vanilla policy gradient method [50], which usually samples inefficient and suffers from the high variance of $R_t$. Actor-critic methods [51] additionally learns a model of $V$ or $Q$ as a "critic" to the "actor" $\pi$, which performs as a baseline to effectively reduce the variance. The optimization of $V$ or $Q$ aims to minimize the Bellman residual

$$J_{Q^\pi} = \mathbb{E}_{(s_t, a_t, g)} \left[Q^\pi(s_t, a_t|g) - r(s_t, a_t|g) - \gamma \mathbb{E}_{s_{t+1}}[V(s_{t+1}|g)]\right]^2, \tag{1}$$

Given the critic $Q$, maximizing the expected return w.r.t. $\pi$ reduces to minimizing

$$J_\pi = \mathbb{E}_{(s_t, g)}[-V(s|g)] = \mathbb{E}_{(s_t, g)}[\mathbb{E}_{a_t}[-Q^\pi(s, a|g)]]. \tag{2}$$

A typical actor-critic algorithm alternates between minimizing $J_Q$ and $J_\pi$. To encourage exploration, we use soft actor-critic (SAC) [20] that augments $V$ with an entropy term (with temperature $\alpha$), i.e.,

$$V(s|g) = \mathbb{E}_{a \sim \pi}[Q^\pi(s, a|g) - \alpha \log \pi(a|s, g)]. \tag{3}$$

In order to encourage the above equation, SAC additionally optimizes $V$ by minimizing the mean square error

$$J_V = \mathbb{E}_{(s_t, g)} \left[V(s_t|g) - \mathbb{E}_{a \sim \pi}[Q^\pi(s_t, a_t|g) - \alpha \log \pi(a_t|s_t, g)]\right]^2. \tag{4}$$

SAC alternatively optimizes $J_V$, $J_Q$ and $J_\pi$ (using the augmented $V$ in Eq. (3)) defined in Eq. (1)-(4) by using stochastic gradient decent (SGD) on batches of sampled $(s_t, a_t, g)$. Although we use SAC in our experiments, CO-PILOT can work with any other RL algorithm besides SAC.

## 3.2 Reward Shaping for RL by Path-Planning

In various environments, an RL agent receives a nonzero reward only when reaching the $\epsilon$-ball $B(g, \epsilon)$ around the goal $g$, i.e., $r(s, a|g) = \mathbb{1}[s \in B(g, \epsilon)]$ with $\mathbb{1}$ being the indicator, so $r(s, a|g)$ for most steps/trajectories cannot provide informative feedback to policy training. RL is unstable and can easily fail with such sparse reward, especially in long-horizon tasks when $g$ is far away from $s_0$ or too difficult to reach for the agent-in-training. To address this problem, reward shaping method [31] augments the environment reward with a dense reward $r'(s, a|g)$ that can be issued to more non-goal states, e.g., intrinsic motivation/curiosity that encourages effective exploration, or human-engineered task-specific rewards. The ideal dense reward, which is, however unavailable without knowing $\pi^*$, is $V^*(s|g)$. Planning methods, e.g., value iteration [52] or fitted-Q iteration [4], can approximate $V^*(s|g)$ but accurately estimating $V^*(s|g)$ is as challenging as the policy learning.

Path-planning and motion-planning methods [12] usually adopt a heuristic distance or cost $c(s, g)$ (e.g., Euclidean distance or time cost) to replace the unknown $V^*(s|g)$. They discretize the state space into a grid/graph and find the shortest path connecting the initial state $s_0$ and the goal $g$. In goal-conditioned RL, $s_0$ and $g$ can be any feasible states on the graph, so path-planning needs to solve the all-pairs shortest path(APSP) problem [43], i.e.,

$$\min_{g_0 = s_0, g_{1:T-1}, g_T = g} \sum_{t=0}^{T-1} c(g_t, g_{t+1}), \quad \forall s_0 \in \mathcal{S}, g \in \mathcal{G}, \tag{5}$$

where $g_{1:T-1}$ denotes a discrete sequence of sub-goals $(g_1, g_2, \ldots, g_{T-1})$ between $g_0 = s_0$ and $g_T = g$. Planning every step for an agent is usually challenging and requires an accurate $c(\cdot, \cdot)$ or

environment model. But we only use planning for reward shaping, which can tolerate a small $T$, i.e.,

$$\bar{r}(s, a|g_{0:T}) = \frac{1}{T} \sum_{t=1}^{T} r(s, a|g_t). \tag{6}$$

As $T$ increases, $\bar{r}(s, a|g_{0:T})$ becomes denser and the RL agent can receive more effective feedback for policy training. However, the quality of $\bar{r}(s, a|g_{0:T})$ also heavily depends on the cost for the agent to reach $g$ by following the path $g_{0:T}$, since issuing reward to an inefficient/long path misleads the policy training. Therefore, in CO-PILOT, we use a prediction model to predict $c(\cdot, \cdot)$ and train the planning policy to generate $g_{0:T}$ incurring the smallest cost as in Eq. (5).

**Remarks:** This leads to a collaborative learning and mutual boosting scheme between RL and planning: planning produces easy sub-tasks that the RL agent can complete in a few steps and thus provide dense rewards enabling more efficient RL , while the time costs of RL agent on those sub-tasks can be used to further improve the planning policy towards producing more cost-efficient paths and better reward shaping. In addition, this scheme makes both RL and planning easier to overcome their bottlenecks: RL **learns from dense rewards to complete long-horizon tasks**, while relatively coarse (with small $T$) planning suffices to provide dense rewards so every-step planning relying on accurate modeling of MDP is not necessary. We will introduce more details next.

### 3.3 Curriculum for Learning to Plan a Sub-task Tree

In the above scheme, planning serves RL like a copilot in an aircraft to encourage more efficient training. The main advantage of a tree structure planner is to provide a global view of future milestones to the RL policy, which mainly focuses on local steps and might lack long-term sight. However, many planning algorithms are based on Bellman equation and sequentially predict the sub-goals, which may suffer from accumulated errors [42]. In addition, as the aforementioned, a larger $T$ results in easier sub-tasks for the RL agent but also increases the difficulty of planning, and vice versa. Hence, it is challenging to train both the RL and planning policy from scratch using either a small or a large $T$. This motivates us to seek coarse-to-fine planning that can generate multiple trajectories of sub-goals with increasing $T$, so the planning policy can be trained on an easy-to-hard curriculum [5, 14], i.e., generating coarse-to-fine shortest paths from small $T$ to large $T$. At the same time, the RL agent can also be trained on an easy-to-hard curriculum of sub-tasks, i.e., by following the trajectories from large $T$ to small $T$.

Therefore, we apply "sub-goal tree (SGT)" [25] to recursively divide a trajectory from small $T$ to large $T$ and produce a sub-task tree. We define a planning policy $\pi_p(g|g_i, g_j)$ as a stochastic mapping from two nearby endpoints $g_i$ and $g_j$ to a predicted sub-goal $g$ in the middle of $g_i$ and $g_j$. In our scheme, we use $\pi_p(g|g_i, g_j)$ to break down a task with initial state $g_i$ and goal $g_j$ (denoted by $(g_i, g_j)$) to two sub-tasks $(g_i, g)$ and $(g, g_j)$. Hence, we can generate a tree of sub-goals by recursively sampling sub-goals from $\pi_p(g|g_i, g_j)$ as below, which finally generates a planning trajectory $g_{0:T}$ with a tree structure, i.e.,

$$\Pr_{\pi_p}(g_{0:T}|g_0 = s_0, g_T = g) \triangleq \Pr_{\pi_p}\left(g_{0:\frac{T}{2}} \Big| g_0, g_{\frac{T}{2}}\right) \Pr_{\pi_p}\left(g_{\frac{T}{2}:T} \Big| g_{\frac{T}{2}}, g\right) \pi_p\left(g_{\frac{T}{2}} \Big| s_0, g\right), \tag{7}$$

where $T = 2^K$ with $K$ being the depth of the tree. As shown in Figure 1 (a), for layer-$k$, the sub-goal tree $g_{0:T}$ interpolates a sequence of $2^k - 1$ sub-goals $g_{1:(2^k-1)}^k \triangleq \left(g_1^k, g_2^k, \ldots, g_{2^k-1}^k\right)$ between $s_0$ and $g$, where $g_j^k = g_{Tj/2^k}$ in $g_{0:T}$, $\forall j \in [2^k-1]$. In layer-1, we have the coarsest trajectory $(s_0, g_1^1 = g_{T/2}, g)$. In the bottom layer-$K$, we have the finest trajectory $g_{0:T}$. From top layers to bottome ones, their sub-goal trajectories naturally form a coarse-to-fine sub-tasking curriculum, e.g., the planning in layer-1 requires the agent to accomplish two hard and long-horizon sub-tasks to reach $g$, while layer-$K$'s planning requires the agent to accomplish $T$ much simpler and shorter-horizon sub-tasks.

**To train the planning policy** $\pi_p$, we apply it to produce a tree-structured $g_{0:T}$ via Eq. (7) and evaluate the cost $c\left(g_{0:2^k}^k\right)$ of the trajectory $g_{0:2^k}^k$ by integrating the cost of every segment/sub-task $c\left(g_{tT/2^k}^k, g_{(t+1)T/2^k}^k\right)$ along the trajectory, i.e., $c\left(g_{0:2^k}^k\right) \triangleq \sum_{t=0}^{2^k-1} c\left(g_{tT/2^k}^k, g_{(t+1)T/2^k}^k\right)$. We will elaborate on our option of cost function $c(\cdot, \cdot)$ later in Eq. (11). The objective of $\pi_p$ aims to minimize the total cost $c(g_{0:T})$ of the sub-goal tree, which sums over all trajectories' costs across the $K$ layers,

$$c(g_{0:T}) \triangleq \sum_{k=0}^{K} c\left(g_{0:2^k}^k\right). \tag{8}$$

According to APSP objective in Eq. (5), the optimal planning policy $\pi_p^*$ minimizes the expected cost $J_{\pi_p}$ over all possible planning trajectories defined below:

$$J_{\pi_p} \triangleq \mathbb{E}_{g_{0:T}}[c(g_{0:T})] = \mathbb{E}_{(s_0,g)}\mathbb{E}_{g_{1:T-1}\sim\pi_p}[c(g_{0:T})], \tag{9}$$

where $g_{1:T-1} \sim \pi_p$ denotes the recursive sampling of $g_{1:T-1}$ in Eq. (7). Any policy gradient method can be used to minimize $J_{\pi_p}$, with the gradient w.r.t. $\pi_p$ computed as

$$\nabla J_{\pi_p} = \mathbb{E}_{g_{0:T}\sim\pi_p}\left[c(g_{0:T}) \cdot \nabla \log \Pr_{\pi_p}(g_{0:T}|s_0,g)\right]. \tag{10}$$

To form an easy-to-hard curriculum for training $\pi_p$, during the top-down construction of the tree, at every layer-$k$, we train $\pi_p$ to only minimize the cost for layers from $0$ to $k$ (instead of $K$ as in Eq. (8)) so the planning policy $\pi_p$ starts from only producing relatively coarse sequence of a few sub-tasks for the top layers before trained to produce more detailed sub-task paths. In Line 4 of Algorithm 1, we will use $J_{\pi_p}^k$ to denote $c(g_{0:T})$ computed up to layer-$k$.

**Cost function of sub-tasks:** As discussed in the end of Section 3.2, the cost function $c(g_t, g_{t+1})$ should reflect the difficulty of sub-task $(g_t, g_{t+1})$ for the RL agent. Euclidean distance $\|g_t - g_{t+1}\|_2$ is commonly used by previous path-planning methods but is not adaptive to the evolution of the agent's policy and environment, e.g., the difficulty of sub-task $(g_t, g_{t+1})$ with/without nearby obstacles can vary drastically. Instead, we use the time-cost $\tau_{g,g'}$ spent by the agent on completing the task $(g, g')$ to measure its difficulty, which is adaptive to both the agent and environment and thus more accurate than Euclidean distance. By training the planner to produce minimum-cost sequences of sub-tasks, the planned paths are optimized for the training of RL policy. Since the time cost data is collected during the training of RL agent on the assigned sub-tasks, they are not available at the very beginning of the first episode. Therefore, we "warm start" the first top-down construction and training of the planner by Euclidean distance and consider the following cost function for the first two episodes, i.e.,

$$c(g_t, g_{t+1}) = \alpha\|g_t - g_{t+1}\|_2 + (1-\alpha)\tau(g_t, g_{t+1}). \tag{11}$$

In experiments, we start from $\alpha = 1$ and gradually reduce it towards $0$ during the first two episodes. After that, the cost function is the time cost $\tau(g_t, g_{t+1})$ collected in previous episode and no longer depends on the Euclidean distance.

## 3.4 Mutual Training between RL and "Learning to Plan"

CO-PILOT is a mutual training scheme between the RL policy $\pi$ and the planning policy $\pi_p$, each generating dense cost/reward on tree-structured sub-tasks to train the other.**By top-down construction of sub-task tree from $k = 0$ to $k = K$,** it firstly trains the planning policy $\pi_p$ on a curriculum of generating coarse-to-fine trajectories. On each layer-$k$, it generates $2^k$ sub-tasks through rollouts of $\pi_p$. Given $\mathcal{D}_\tau$ and the cost

---

**Algorithm 1** Top-Down Construction of Sub-Task Tree

1: **Input:** $(s_0, g)$, planning policy $\pi_p$ and its training set $\mathcal{D}_\tau$
2: **Output:** tree structured sub-goals $g_{0:T}$, $\pi_p$
3: **for** $k = 1, 2, \ldots, K$ **do**
4:     Apply any RL method to minimize $J_{\pi_p}^k$, i.e., $J_{\pi_p}$ in Eq. (9) computed only up to layer-$k$;
5:     **for** $t = 0, 1, \ldots, 2^{k-1} - 1$ **do**
6:         Generate the sub-goal $g_t^k \sim \pi_p(g_t^k|g_{t-1}^{k-1}, g_t^{k-1})$;
7:         Add $g_t^k, g_t^{k-1}$ into the trajectory $g_{0:T}^k$ on layer-$k$;
8:     **end for**
9: **end for**

---

$c(\cdot, \cdot)$ in Eq. (11), CO-PILOT updates $\pi_p$ by minimizing $J_{\pi_p}$ in Eq. (9). At the very beginning of CO-PILOT, $\mathcal{D}_\tau = \emptyset$ and the cost solely depends on the Euclidean distance. However, as we are collecting more experiences into $\mathcal{D}_\tau$, $\pi_p$ will be trained towards producing the easiest sub-task trajectory for the RL agent to finish and thus increases its chance of receiving non-zero rewards. The complete procedures of top-down construction are given in Algorithm 1. Being updated using the most recent $\mathcal{D}_\tau$, $\pi_p$ keeps tracking the RL agent's learning progress to produce the most cost-efficient paths for the agent. Moreover, the top-down construction naturally forms an **easy-to-hard curriculum for the planning policy** $\pi_p$. In Line 4 of Algorithm 1 , we train $\pi_p$ to produce sub-goal trajectories up to layer-$k$. Hence, the training of $\pi_p$ is more smooth and less challenging than learning the optimal $V^*$ or $Q^*$.

After the top-down construction of the sub-task tree, CO-PILOT trains the **RL policy $\pi$ on a curriculum of easy-to-hard** sub-tasks by **bottom-up traversal** of the tree from $k = K$ to $k = 0$. The sub-goal trajectory in each layer aims to guild the agent to complete the original task from $s_0$ to $g$. At layer-$k$, CO-PILOT applies $\pi$ sequentially to the $2^k$ sub-tasks (as the conditioned goal). It then updates $\pi$ by SAC, which alternates among the minimization of $J_Q$, $J_\pi$ and $J_V$ in Eq. (1)-(4) to update $\pi$, $V$ and $Q$.

Note we can replace SAC with other RL algorithms in the general framework of CO-PILOT. The rollouts of $\pi$ on the sub-tasks not only collect experiences to train itself but also collect tuples of $(g, g', \tau_{g,g'})$ added to $\mathcal{D}_\tau$, which will be used to train $\pi_p$. It is possible that $\pi$ fail on some sub-task within $\tau_{\max}$ steps. In this case, we treat the actual ending state as $g'$ in the tuple for $D\tau$ and initialize the the next sub-task from this state. The bottom-up traversal is detailed in Algorithm 2, where Line 9-13 apply $\pi$ to reach sub-goal $g_{tT/2^k}$. The bottom-up traversal forms an easy-to-hard curriculum to train $\pi$, in which the sub-tasks from the bottom layers are easier so the agent by larger chance can receive non-zero rewards. Given a task $(s_0, g)$, the curriculum guides the agent first to learn how to finish it by following a detailed

---

**Algorithm 2** Bottom-Up Traversal of Sub-Task Tree

1: **Input:** RL policy $\pi$, sub-goal tree of $g_{0:T}$, $\tau_{\max}$, $\epsilon$
2: **Output:** $\pi$, $\mathcal{D}_\tau$
3: **Initialize:** $\mathcal{D}_\tau \leftarrow \emptyset$
4: **for** $k = K, \ldots, 1, 0$ **do**
5:     Set RL agent's initial state to be $s_0 \leftarrow g_0$;
6:     **for** $t = 1, 2, 3, \ldots, 2^k$ **do**
7:         Set the condition of $V, Q, \pi$ in SAC to be $g_t^k$;
8:         $\tau \leftarrow 0$,  $\mathcal{B} \leftarrow \emptyset$;
9:         **while** $\tau \leq \tau_{\max}$ or $s_\tau \notin B(g_t^k, \epsilon)$ **do**
10:           RL agent takes action $a_\tau \sim \pi(a_\tau | s_\tau, g_t^k)$;
11:           RL agent moves to $s_{\tau+1} \sim p(s_{\tau+1} | s_\tau, a_\tau)$ and receives reward $r(s_\tau, a_\tau | g_t^k)$;
12:           $\mathcal{B} \leftarrow \mathcal{B} \cup (s_\tau, a_\tau, r(s_\tau, a_\tau | g_t^k), s_{\tau+1})$;
13:         **end while**
14:         $\mathcal{D}_\tau \leftarrow \mathcal{D}_\tau \cup (s_0, s_\tau, \tau)$,  $s_0 \leftarrow s_\tau$;
15:         **for** every gradient step **do**
16:           Apply gradient steps in SAC: update $Q, V, \pi$ to minimize $J_Q, J_\pi$ and $J_V$ in Eq. (1)-(4) using samples drawn from $\mathcal{B}$;
17:         **end for**
18:     **end for**
19: **end for**

---

planning path of many sub-goals. It then gradually increases the hardness by halving the number of sub-goals until recovering the original task. Therefore, it critically alleviates the sparse reward problem that usually fails or considerably slows down RL on long-horizon tasks.

A prominent advantage and difference of CO-PILOT compared to other methods that combine RL and planning is to repeat the top-down construction and bottom-up traversal for multiple ($b$ in Algorithm 3) episodes on each task $(s_0, g)$. Thereby, the RL agent and the planning policy are fully optimized for each other's training, forming an adaptive curriculum without human engineering. The complete procedures of CO-PILOT are listed in Algorithm 3.

---

**Algorithm 3** CO-PILOT

1: **Input:** $\mathcal{G}, p_0, T, \tau_{\max}, \epsilon, b$
2: **Output:** RL agent's policy $\pi$, planning policy $\pi_p$
3: **Initialize:** $\pi, \pi_p, \mathcal{D}_\tau$ by Euclidean distance
4: **while** not converge **do**
5:     Sample a task $(s_0, g)$ with $s_0 \sim p_0(s)$ and $g \in \mathcal{G}$;
6:     **for** episode $= 1, 2, \ldots, b$ **do**
7:         **Algorithm 1**: top-down construction of a sub-task tree $g_{0:T}$, train planning policy $\pi_p$ based on $\mathcal{D}_\tau$;
8:         **Algorithm 2**: bottom-up traversal of the sub-task tree $g_{0:T}$, train RL policy $\pi$, collect $\mathcal{D}_\tau$;
9:     **end for**
10: **end while**

---

# 4 Experiments

We evaluate CO-PILOT on three types of tasks: a maze and two continuous control tasks for robotic navigation. CO-PILOT outperforms several strong baselines of RL and planning, as well as methods combining RL and planning, on both sample efficiency and final success rate.

**Maze environment:** We build a maze environment of size $1 \times 1$ containing square obstacles(obstacle states) and free-to-reach states as shown in Figure 4. For each benchmark(the design of benchmark refer to the caption of Figure 2), We randomly sample 300 pairs of $(s_0, g)$ for training and 100 pairs for test from a uniform distribution on the coordinate range and remove the ones in obstacles. It takes an RL agent $\geq 200$ steps on average moving from $s_0$ to $g$, which is a long-horizon task. The task succeeds if the agent reaches $B(g, \epsilon = 0.025)$ without collision.

**Mujoco Ant-v1:** We evaluate CO-PILOT and baselines in the Mujoco environment with an Ant-v1 agent (Quadruped [53]) with an 8-dim action space. As shown in Figure 2 (c), we train the agent to navigate in the maze without self-rotation and collisions to the wall. We randomly sample 50 $(s_0, g)$ pairs for training and 10 pairs for test.

**BipedalWalker:** The BipedalWalker environment [6] offers a new perspective of tasks rather than maze type. The learning agent, embodied in a bipedal walker, receives positive rewards for moving forward and penalties for torque usage and angular head movements. Agents are allowed 2000 steps to reach the other side of the map. The environment producing tracks paved with stumps varying by their height parameter $\mu_h$ and an independent sampled spacing parameter $\Delta_s \in \mathcal{N}(\mu_h, 0.1)$. We design three agents with different length of legs (as shown in figure 3) for each benchmark. Performance is evaluated periodically by sampling 10 tracks in each track distribution of a fixed evaluation set of 50 distributions sampled uniformly in the parameter space. We measure the percentage of mastered tracks.

**Baselines:** In the maze environment, we compare CO-PILOT with (1) three planning methods: RRT* [28] (Rapidly-exploring Random Trees), NEXT [7] (Neural Exploration-Exploitation Trees) and SGTPG [25] (Sub-Goal Tree Policy Gradient); three model-free RL algorithms: valued-based method SAC [21], policy-based method PPO [47] (in CO-PILOT, we use the former to train the RL policy and the latter to train the planning policy) and HER [3], which improves goal-conditioned RL's efficiency by re-labelling the visited states as pseudo goals; and (3) a RL-planning hybrid method: SoRB [13], which trains planning strategies based on the experiences of a given RL policy. For fair comparisons, we use SAC as the RL algorithm in both CO-PILOT and SoRB. In the Mujoco environment, we compare CO-PILOT with SAC, SoRB, and hierarchical RL [23]. In BipedalWalker, we compare CO-PILOT with SAC and SoRB.

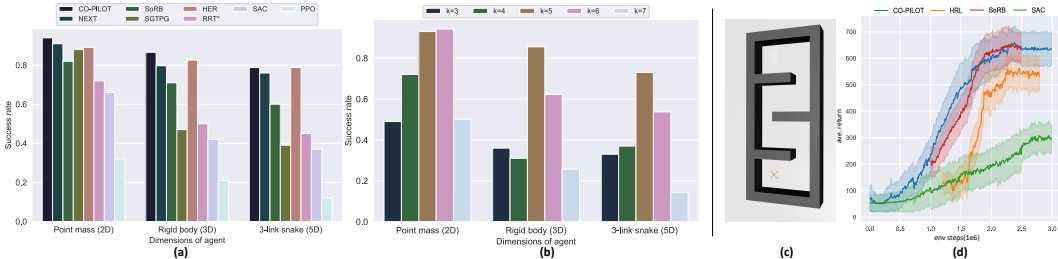

Figure 2: **(a)** Success rate on test tasks of Maze environment. We train three types of agents with different DoF (degrees of freedom): point mass (2D), rigid body (3D), and 3-link snake (5D). More details are given in Appendix B. **(b)** Success rate of CO-PILOT with sub-goal tree of different depth $K$ on the same test set in **(a)**. **(c)** Ant-v1 agent and the associated environment in Mujoco. **(d)** Average return of Ant-v1 in **(c)**.

## 4.1 Training Details and Hyperparameters

In CO-PILOT, we initialize the dataset $\mathcal{D}_\tau$ with 50,000 tuples of $(g, g', \tau_{g,g'})$ with $\tau_{g,g'}$ being the Euclidean distance. We use SAC to train the RL policy and PPO [47] to train the planning policy in Line 8 of Algorithm 1, since the former encourages exploration and the latter is simple and efficient. We set a reward of 1 (1000, 200) to each task $(s_0, g)$ in Maze (Mujoco,BipedalWalker). According to Eq. (6), the reward of each sub-task in layer-$k$ is $1/2^k$ ($1000/2^k, 200/2^k$). For planning cost, if the segment between $(g_t, g_{t+1})$ trespasses any obstacle, we add a penalty of 10 to $\tau(g_t, g_{t+1})$ in Eq. (11). We linearly reduce $\alpha$ in Eq. (11) from 0.9 to 0.1 throughout every episode. For planning policy training, we apply PPO with a trust region of $\epsilon = 0.2$ and use Adam optimizer [30] with a learning rate of 0.005. For RL training with SAC, we use its default hyperparameters. In both environment, we set $T = 2^5$ (ablation study of different $T$ in Figure 2 (c)) and $b = 5$ (further increasing it does not improve the performance). We set $\tau_{\max} = 25, \tau_{\max} = 200$ and $\tau_{max} = 2000$ for Maze, Mujoco and BipedalWalker respectively. For efficiency, in Line 5 of Algorithm 3, we instead sample a mini-batch of 30 (Maze) or 50 (Mujoco) pairs of $(s_0, g)$. The $(s_0, g)$ is fixed in BipedalWalker, we randomly sample 20 tracks in each track distribution from the same 50 distributions mentioned in section 4.1.

## 4.2 Main Results

In Figure 2 (a), we compare the performance of CO-PILOT with all the baselines on the test tasks of the Maze environment. CO-PILOT achieves the highest success rate across all benchmarks and significantly outperforms SAC and SGTPG. Figure 4 (a)-(c) report how the success rate of all methods change during training as the number of interaction steps with the environment increases. We limit the total environment steps of all methods $\leq 1.8 \times 10^6$ except for NEXT (since NEXT

requires the pre-training of RRT*). In Figure 4 (a), CO-PILOT and SGTPG perform similarly in the early training period because the cost data collected by the RL agent do not contain sufficient information to train a powerful path-planner and the inaccurate Euclidean distance dominates the cost $c(\cdot, \cdot)$ in Eq. (11). The performance of SoRB and SAC are similar because SoRB needs to pre-train the RL policy at first when no planning is required. SoRB surpasses SAC during the later stages. NEXT also needs to pre-train RRT* before applying self-improving training, so we do not see the change of cost and collision checks for NEXT during the earlier stages. The comparison between CO-PILOT with SAC demonstrates the significant improvement brought by the learned planner to RL.

The average return on Mujoco tasks is shown in Figure 2 (d). Furthermore, the percentage of mastered environments on BipedalWalker is shown in Figure 3. For simplicity, we denote [23] as HRL. SoRB and HRL start later because the RL policy is under pre-training. The experimental results show that CO-PILOT achieves much better sample efficiency than all the baselines, including SoRB. The final performance is comparable with SoRB but significantly outperforms SAC and HRL(Mujoco).

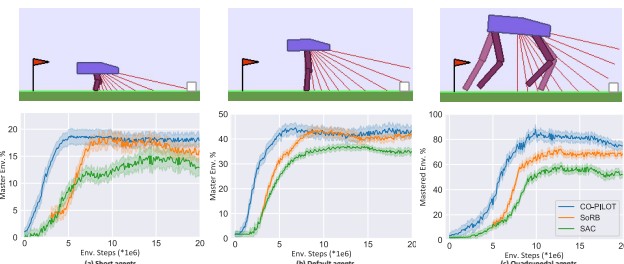

Figure 3: **Main results and comparison in BipedalWalker.** The mean performance (32 seeded runs) is reported together with the standard deviation (shaded areas).

**CO-PILOT with sub-task tree of different depth $K$:** In Figure 2 (b), we evaluate 5 different depths for the sub-task tree in CO-PILOT, with all the rest hyperparameters fixed. It shows a trade-off between RL and planning, i.e., a deeper sub-task tree can provide denser rewards and more detailed guidance from sub-tasks, hence improving the efficiency of RL, but it also makes training the planning policy more challenging. In this experiment, the best trade-off is achieved when depth $K = 5$.

## 4.3 A Close Look of Mutual Training in CO-PILOT

To understand the mutual training of CO-PILOT in the experiments, in Figure 4 (d), we visualize how a sub-task tree evolves over episodes in Algorithm 3, where the sub-goal paths at layer-2, 3, 4 generated in Episode-1, 3, 5 for the same task $(s_0, g)$ are reported. Each maze map contains a path from a layer in an episode and the histogram above it reports the time cost of RL for completing each sub-task on the path. We are particularly interested in two questions: how does planning guide RL by sub-tasking? How does RL agent's cost affect the planned paths?

**Train $\pi_p$ using RL agent's time cost data.** In Episode-1, the sub-goal paths of all the three layers are too close to some obstacles or even trespass some others and thus cannot provide reliable guidance for the RL agent. The Euclidean distances between consecutive sub-tasks on a path are almost equal but the corresponding time costs shown in the histograms vary a lot, which is not preferred since some sub-tasks are too hard, but some are too easy for training the RL agent. Hence, the planning policy is not fully optimized to produce cost-efficient paths for the RL agent.

In Episode-3, the generated sub-goals paths become more adaptive to the environment. In all layers, we can see that the planner tends to generate longer segments for places with fewer nearby obstacles and collision risks while adding more fine-grained sub-tasks to get around the corners. This phenomenon implies that the planning policy is learning to produce better and more adaptive guidance with dense rewards. However, due to the limited number of sub-goals per layer, the paths in layer-2, 3 can still be improved if interpolating more sub-goals. Nevertheless, on the deepest layer-3, the planned path is already collision-free and thus can provide an accurate reward shaping for the RL agent.

In Episode-5, the planning paths are almost optimal, especially for the one in layer-4, which keeps distant from the obstacles of both sides in the maze. Moreover, RL agent's time costs shown in the histograms are not only much lower than those of the previous two episodes but also have similar values across different sub-tasks. Hence, the planning policy is well optimized to generate cost-efficient paths for RL.

**Planner guides RL by sub-tasking.** In Episode-1, the time cost for the RL agent to finish the whole task is much higher than that in the later episodes due to the poor RL policy at the beginning.

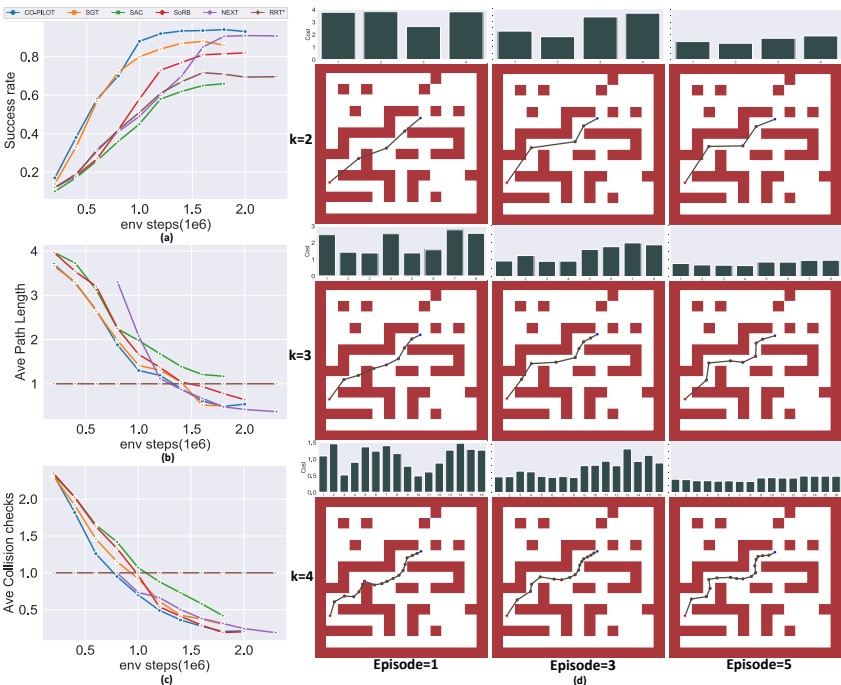

Figure 4: **(a)** Success rate. **(b)** Average path length (normalized by RRT*) in terms of Euclidean distance. **(c)** Average collision checks (normalized by RRT*) as every method increases its interaction steps with the environment. CO-PILOT achieves the best sample efficiency among all methods. **(d)** Visualization of the sub-goal paths on layer-$k = 2, 3, 4$ of the sub-task tree in Episode-1, 3, 5 for a task with initial state $s_0$ (red dot) and goal $g$ (blue dot) in Maze. Each histogram reports the RL agent's cost $\tau_{g_t, g_{t+1}}$ for sub-tasks along the path. As episode increases, planning paths across all layers are improved, and on each path the costs of all sub-tasks reduce towards a similar value, though the Euclidean distances are still different, since the planner learned to produce more sub-tasks near complicated obstacles.

In contrast, the time cost drastically decreases in Episode-3 when compared to Episode-1, which indicates that the RL policy is significantly improved under the guidance of planning. In Episode-5, the time cost for completing each sub-task of layer-3, 4 further decreases when compared to the former episodes and the costs becoming more uniform across sub-tasks, implying that the RL policy learns to complete the planned sub-tasks more efficiently.

Therefore, both planning and RL are improved via the mutual training scheme in CO-PILOT and fully optimized to facilitate the training of the other. In particular, the planner learns to produce cost-efficient paths of different amounts of sub-tasks to guide RL with dense rewards. It does not depends on any pre-defined metric but is adaptive to the RL policy. Moreover, it does not need to produce a step-by-step plan: the RL agent can learn efficiency under the guidance of a few interpolated sub-tasks by the planner. Hence, the mutual training between the two policies overcome the bottlenecks of training each policy separately. In addition, the easy-to-hard curricula for both planning and RL considerably improves their training efficiency.

## 5   Conclusion

We propose CO-PILOT, a mutual learning framework between RL and "learning to plan" policies, which provides a principal solution addressing the problems of both RL and planning when applied to long-horizon tasks. In CO-PILOT, each policy produces dense feedback on a curriculum of sub-tasks to train the other more efficiently and is optimized to assist the other's training. The planner learns to decompose a long-horizon task into a few sub-tasks at first and then gradually increases the interpolated sub-tasks, forming an easy-to-hard curriculum to train the planning policy. On the other hand, this top-down curriculum recursively builds coarse-to-fine sequences of sub-tasks. By training the RL agent to complete easier sub-tasks on finer sequences of bottom layers at first and then gradually moving to harder ones in top layers, the RL agent can be efficiently trained following an easy-to-hard curriculum. In experiments, CO-PILOT significantly improves the sample efficiency and success rate on different types of tasks especially on long-horizon tasks with sparse rewards.

## Acknowledgments

This work was supported by Australian Research Council (ARC) LP180100654 and CSIRO/DATA61 CRP Project C019309. We would like to thank NeurIPS area chairs and anonymous reviewers for their efforts in reviewing this paper and their constructive comments!

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
