# A Policy Gradient for Sub-task Tree

## A.1 Recursive evolution

Formally, assume an initial pair $(s_0, g)$ sampled, which defined the root of the tree. Next, a policy $\pi(s_{T/2}|s_0, g)$ is used to predict the sub-goal $s_{T/2}$ segmenting $(s_0, g)$ into $(s_0, s_{T/2})$ and $(s_{T/2}, g)$. Recursively each segment is again partitioned by $\pi$. Then results in Eq. (7):

$$\Pr_{\pi_p}(g_{0:T}|g_0 = s_0, g_T = g) \triangleq$$

$$\Pr_{\pi_p}\left(g_{0:\frac{T}{2}}\left|g_0, g_{\frac{T}{2}}\right.\right) \Pr_{\pi_p}\left(g_{\frac{T}{2}:T}\left|g_{\frac{T}{2}}, g\right.\right) \pi_p\left(g_{\frac{T}{2}}\left|s_0, g\right.\right).$$

For example consider a sub-task tree with K= 3, given a $(s_0, g)$ pair, the sub-tasks will be $s_1, ..., s_7$. Then, we have:

**k=1**, the trajectory of planned sub-goals.

$$s_0^1 = s_0,$$
$$g_1^1 = \pi_p(s_0, g|s_0, g),$$
$$g_2^1 = g.$$

Then, the whole planned sub-goals trajectory in layer k= 1, $\tau_{k=1} = \left\{s_0, g_1^1, g_2^1 = g\right\}$.

**k=2**, in this case, $t \in [1, 2^2 = 4]$.

$$s_0^2 = s_0^1 = s_0, \quad g_1^2 = \pi_p(s_0, g_1^1|s_0, g),$$
$$g_2^2 = g_1^1, \quad g_3^2 = \pi_p(g_1^1, g_2^1|s_0, g),$$
$$g_4^2 = g.$$

Then, the whole planned sub-goals trajectory in layer k= 2, $\tau_{k=2} = \left\{s_0, g_1^2, g_2^2 = g_1^1, g_3^2, g_4^2 = g\right\}$.

**k=3**, in this case, $t \in [1, 2^3 = 8]$.

$$s_0^3 = s_0^2 = s_0, \qquad g_1^3 = \pi_p(s_0, g_1^2|s_0, g), \quad g_2^3 = g_1^2,$$
$$g_3^3 = \pi_p(g_1^2, g_2^2|s_0, g), \quad g_4^3 = g_2^2, \qquad\qquad g_5^3 = \pi_p(g_2^2, g_3^2|s_0, g),$$
$$g_6^3 = g_3^2, \qquad\qquad g_7^3 = \pi_p(g_3^2, g_4^2|s_0, g), \quad g_8^3 = g.$$

Then, the whole planned sub-goals trajectory in layer k= 3, $\tau_{k=3} = \left\{s_0, g_1^3, g_2^3 = g_1^2, g_3^3, g_4^3 = g_2^2, g_5^3, g_6^3 = g_3^2, g_7^3, g_8^3 = g\right\}$.

## A.2 Proof of Eq. (10)

**Proposition 1.** Assuming a planning policy $\pi_p$ with parameters $\theta$, we now prove a policy gradient for computing $\nabla J_{\pi_p}$, and $K = k_{max}, T = 2^K$. Then the cost of trajectory can be written as:

$$\nabla \log Pr[g_{0:T}|g_0 = s_0, g_T = g] =$$

$$\sum_{k=1}^{K} \sum_{t=1}^{2^k} \nabla \log \pi_p\left(g_{(2t+1)\cdot 2^{K-k}}^k \left| g_{t\cdot 2^{K-k+1}}^k, g_{(t+1)\cdot 2^{K-k+1}}^k\right.\right).$$

*Proof.* First, we express $Pr[g_{0:T}]$ as,

$$Pr[g_{0:T}|g_0 = s_0, g_T = g] = \rho(g_t, g_{t+1})\cdot$$

$$\prod_{k=1}^{K} \prod_{t=1}^{2^k} \pi_p\left(g_{(2t+1)\cdot 2^{K-k}}^k \left| g_{t\cdot 2^{K-k+1}}^k, g_{(t+1)\cdot 2^{K-k+1}}^k\right.\right).$$

Now, by taking log we have,

$$\log Pr[g_{0:T}|g_0 = s_0, g_T = g] = \log \rho(g_t, g_{t+1}) +$$

$$\sum_{k=1}^{K} \sum_{t=1}^{2^k} \log \pi_p \left( g_{(2t+1)\cdot 2^{K-k}}^k \,\middle|\, g_{t\cdot 2^{K-k+1}}^k, g_{(t+1)\cdot 2^{K-k+1}}^k \right).$$

The proposition 1 above shows that the gradient of a trajectory does not depend on the initial distribution. This allows us to derive a policy gradient proposition:

**Proposition 2.** Let planning policy $\pi_\theta$ be a stochastic policy, and $T = 2^k$. Then,

$$\nabla J_{\pi_p} = \mathbb{E}_{g_{0:T} \sim \pi_p} \left[ c(g_{0:T}) \cdot \nabla \log \Pr_{\pi_p}(g_{0:T}|s_0, g) \right].$$

To obtain $\nabla J_{\pi_p}$ which is parameterized by $\theta$ we write the Eq. (10) as ax explicit expectation and use $\nabla_x f(x) = f(x) \cdot \nabla_x f(x)$.

$$\nabla_\theta J_{\pi_p} = \sum_\tau c(\tau) \cdot \nabla_\theta \Pr_{\rho(\pi_p)}[\tau]$$

$$= \sum_\tau c(\tau) \cdot \Pr_{\rho(\pi_p)}[\tau] \cdot \nabla_\theta \log \Pr_{\rho(\pi_p)}[\tau]$$

$$= \mathbb{E}_{\rho(\pi_p)} \left[ c(\tau) \nabla_\theta \log \Pr[\tau] \right]$$

The policy gradient theorem allows estimating $\nabla J(\pi_\theta)$ from on policy data collected using $\pi_p$.

## B    Maze Environment Details

We design three benchmark environments in our experiment (as shown in Figure 5). For the first three. We generated the maze maps with the recursive backtracker algorithm using the following implementation:RLAgent. Three environments differ in the choice of robots:

**Workspace planning (2d):** The robot is abstracted with a point mass moving in the plane. Without higher dimensions, this problem reduces to planning in the workspace.

**Rigid body navigation (3d):** A rigid body robot, abstracted as a thin rectangle, is used here. This robot can rotate and move freely without any constraints in the free space.

**3-link snake (5d):** The robot is a 5 DoF snake with two joints.To prevent links from folding, we restrict the angles to the range of $[-\pi/4, \pi/4]$

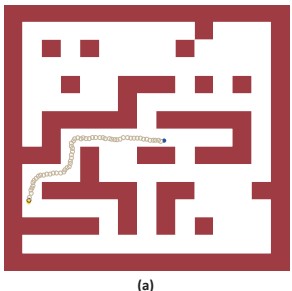 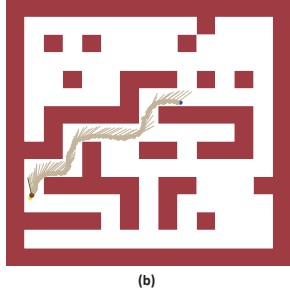 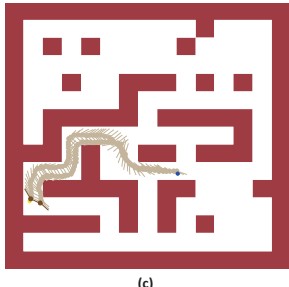

(a)  (b)  (c)

Figure 5: Three different agents in Maze environment. (a) 2d workspace, (b) Rigid body, (c) 3-link snake. The deep brown agent shows when the agent is around the initial state. The light brown agent shows the actual RL agent navigation trajectory. Yellow point and blue point represent for the initial state $s_0$ and goal state $g$ respectively.

## C    Hyperparameter Settings

We list the hyerparameters of SAC in Maze environment and continuous control environment in table 1 and table 2 respectively.

Table 1: Hyperparameter of SAC in Maze environments

| Parameter | Value |
|---|---|
| optimizer | Adam |
| Timesteps | $1.8 \times 10^6$ |
| learning rate | $3 \cdot 10^{-4}$ |
| discount ($\gamma$) | 0.99 |
| replay buffer size | $10^6$ |
| number of hidden layers (all networks) | 2 |
| number of hidden units per layer | 256 |
| number of samples per minibatch | 200 |
| nonlinearity | ReLU |
| target smoothing coefficient ($\tau$) | 0.005 |
| target update interval | 1 |
| gradient steps | 1 |

Table 2: Hyperparameter of SAC in continuous control environments

| Parameter | Value |
|---|---|
| optimizer | Adam |
| Timesteps | $3 \times 10^6$ |
| learning rate | $5 \cdot 10^{-4}$ |
| discount ($\gamma$) | 0.99 |
| replay buffer size | $10^6$ |
| number of hidden layers (all networks) | 2 |
| number of hidden units per layer | 256 |
| number of samples per minibatch | 256 |
| nonlinearity | ReLU |
| target smoothing coefficient ($\tau$) | 0.005 |
| target update interval | 1 |
| gradient steps | 1 |