# OpenReview forum: "CO-PILOT: COllaborative Planning and reInforcement Learning On sub-Task curriculum"
_NeurIPS.cc/2021/Conference — NeurIPS 2021 Poster_

### Official Review · Reviewer_Fa86 · 2021-07-12

**Rating:** 6
**Confidence:** 4

**Summary:**

This paper presents CO-PILOT, a collaborative learning approach in which a planner, generating subtasks of varying complexity, and RL policy, learning to achieve those sub-tasks, are trained together through a curriculum of tree-structured subtasks. This simultaneous curriculum-based training helps to learn functions that can solve a variety of tasks in cluttered environments.

**Limitations And Societal Impact:**

- Limitations of their work need to be highlighted, especially generalization to new environments and how it can be addressed.
- Societal impacts: (N/A)

**Main Review:**

 The paper is well written, and most key aspects of their algorithm are neatly discussed. However, there are some concerns, and some of them can be addressed in the revised version.


1- Related work can be improved:
- Difference between HRL and CO-PILOT should be further explained. The main difference is that the latter introduces a tree-based curriculum for training high and low-level policies, leading to better performance than existing HRL methods.

- Although I see this approach is different from imitation learning-based path planning approaches, including them in the literature would be nice.

2- Methods:
- Fig 1(b) can be described in a bit more detail in the caption.

3- Results:
- Generalization of their approach to different start and goals is not shown. For instance, in the maze, I think 100 (s,g) test pairs should be shown in the appendix to indicate how these samples were distributed, likewise for other environments. In the visuals, it seems the start and goal points were not quite distributed around the space.

- Computational times of different algorithms during execution/testing should be compared, or at least they should be discussed in the paper.

- Also, it is not clear why RRT* has a lower success rate than CO-PILOT. I believe the presented problems are simple enough for RRT* to solve them. Is it due to the time limit?


**Time Spent Reviewing:**

5

---

> ### Author Response · Authors · 2021-08-10
> **Response to  Reviewer Fa86**
>
> Thank you for your time and suggestions! We will improve our related work in the next version. Now, we will address your main concern as below:
>
> ### Q1: Computational times of different algorithms during execution/testing should be compared, or at least they should be discussed in the paper.
>
> - We compare the computational times of CO-PILOT and SoRB during execution for the Maze environment:  to reach a 50% success rate, CO-PILOT needs $\sim$ 22.7 mins while SoRB needs $\sim$29.5 mins. This indicates that the mutual training and “learn to plan” in CO-PILOT do improve the efficiency.
>
> ### Q2: I think 100 (s,g) test pairs should be shown in the appendix to indicate how these samples were distributed, likewise for other environments.
>
> - We will attach the test pairs in the appendix of the next version.To achieve these relatively long-horizon tasks, we uniformly sample (s_0,g) pairs in the state space and then remove the duplicates and short-horizon $(s_0,g)$ pairs whose L1 distance between $s_0$ and $g$ is smaller than half size of the map, i.e., $||s_0-g||_{1}\leq \frac{1}{2}\max\\{w,h\\}$ for a $w\times h$ map. We repeat this process until 400 long-horizon tasks are collected and we apply a random splitting to gain a training set of 300 tasks and a test set of 100 tasks.  As illustrated in L300, each task requires the agent to move at least 200 steps from start point to reach the goal state.
>
> ### Q3: It is not clear why RRT* has a lower success rate than CO-PILOT. I believe the presented problems are simple enough for RRT* to solve them. Is it due to the time limit?
>
> - As shown in Fig. 2(a), most baseline methods including RRT* do not show poor performance on 2D tasks. However, when the agent morphology (detailed in Section B of the Appendix) and the environment become more complex, as shown in Fig. 2(a) and Fig. 4(a), the success rate for some baselines including RRT* drops quickly. Since RRT* uses Euclidean distance as the cost, which is not adaptive to the RL agent and does not take the obstacles into account, this degradation is expected.
>
> ### Q4: Limitations of their work need to be highlighted, especially generalization to new environments and how it can be addressed.
>
> - Although the high-level idea of CO-PILOT is very general, i.e., a planner and an RL agent learn from each other’s feedback on an automatically generated curriculum of tasks, we only evaluated it on two types of tasks, i.e., maze navigation and robotic control. We will evaluate it on more diverse RL tasks and study if it can generally improve the training efficiency. Moreover, due to our limited computational resources, we have not tested it on more complicated tasks, e.g., the ones with dynamic environments or visual inputs such as games. It is still unclear whether it will still bring improvements to these tasks.

---

> > ### Comment · Reviewer_Fa86 · 2021-08-17
> > **Thank you for your response**
> >
> > The authors have addressed my concerns.

---

> > > ### Author Response · Authors · 2021-09-01
> > > **[Last day reminder] Would you mind reconsidering your rating given that all your concerns have been successfully addressed? Thanks!**
> > >
> > > Dear Reviewer Fa86：
> > >
> > > We are glad to hear that all your concerns have been addressed. We humbly expect that you can reconsider your ratings given all these answers. Your support is very important to our work and we greatly appreciate that!
> > >
> > > Best Regards,
> > >
> > > Authors

---

### Official Review · Reviewer_8yHH · 2021-07-17

**Rating:** 6
**Confidence:** 5

**Summary:**

This paper addresses the problem of sequential decision making in complex, high-dimensional domains. The paper combines a learned planning policy and a learned controller to improve the performance of both. Explicit plans are used as reward shaping for the goal-conditions reinforcement learner to improve the learning rate, and the cost of the learned controller policy (“the time steps of the agent to finish a sub-task”) is used to learn a planning policy. The overall approach is compared to planning-only approaches, model-free policy learning, and a previous hybrid technique (SoRB) and the evaluation is on 3 different benchmark problems. CO-PILOT performs the best of the different techniques in both asymptotic performance and speed of learning. In some cases SoRB also has similar asymptotic performance, but requires more data and takes longer to converge to that.


**Limitations And Societal Impact:**

I could not identify a clear statement of the limitations of the paper anywhere in the paper. The word “limitation” does not appear in the paper. It is possible that the authors intended that the trade-offs between ease of learning vs complexity of planning as a function of sub-task tree depth (section 4.3, final paragraph) to be an assessment of the limitations, but this is not particularly useful, and is not a statement of the limitation of the technique — what would be more useful is analysis of the kinds of problems where the technique fails. Under what conditions is the planner unable to guide the learner, and another technique should be used?

The author checklist indicated that the authors did not discuss any potential negative societal impacts of the work.


**Main Review:**

This paper proposes an interesting idea. The core insight that “planning at even a coarse level can be used for reward shaping and substantially improves RL on long-horizon tasks with sparse reward” is not especially novel and would seem to be central insight of a considerable body of work in hierarchical abstractions for sequential decision making. However, the idea that the planner and learner provide training signals to each other is interesting and potentially novel. I am moderately predisposed to recommend this paper for acceptance, but it is hard to argue strongly for it, in that the narrative is really not clearly articulated, and the experiments could still use further improvement.

My primary concern is that the paper is sitting in a *very* crowded intersection of competing ideas, and does not really engage with the substantial literature in learning hierarchical abstractions. There are some references, so the authors seem to know about some work in hierarchical abstractions, although many more should be added, such as recent work by Konidaris (“From Skills to Symbols”, 2018) and older work such as Pineau et al (“Policy-contingent abstraction for robust robot control”, 2003), Michini and How (““Bayesian nonparametric inverse reinforcement learning”, 2012) and Surana and Srivastava (“Bayesian nonparametric inverse reinforcement learning for switched markov decision processes”, 2014). That being said, more experimental baseline comparisons are essentially, especially to hierarchical representations.

Additionally, the paper needs to more strongly articulate what is different about their approach, and what specific benefits are enabled. For instance, what if the learner were replaced with a fixed set of motion primitives? Figure 4 might suggest that the algorithm is just learning a sampling distribution, and a comparison to “Learning Sampling Distributions for Robot Motion Planning” (Ichter et al, 2018) would be appropriate, especially if learning the controller inside the planning loop really does add something.

The experimental results are ok, but a great deal of the motivation for this technique comes from high-dimensional and long-range planning problems. The Maze and Mujoco Ant-v1 are neither high-dimensional nor long-range (8 dimensions is not high dimensional). Bipedal-Walker is better, but not an especially hard planning problem. I would recommend substantially expanding the scale and dimensional of a Maze-like environment to really show the algorithm learning smart planning strategies and smart controllers.

The paper contains some statements and assumptions about planning that are not quite right.
- “Moreover, planning every step on the path is as difficult as the original RL and requires fine-grained discretization impractical for high-dimensional states.” — a fine-grained discretization is not necessarily required even for high-dimensional states, and random sampling techniques go a long way towards addressing this. Note that random sampling planners are not the same thing as sampling-based search heuristics.
- “Planning only a few milestone states leaves the RL agent to solve relatively long-horizon sub-tasks.” This may be true, but those relatively long-horizon sub-tasks may be extremely easy to solve.
- “Although sampling-based search heuristics can build a graph with a better exploration-exploitation trade-off, they are not optimized for the RL policy, and the graph cannot be shared across tasks.” This is also not true — multi-query planning techniques such as the probabilistic roadmap are explicitly designed for re-use.

I see the comment that this is a resubmission of a previous paper. I appreciate that the authors have gone to the trouble of reporting that, and the changes they made.


**Time Spent Reviewing:**

1.5 hours

---

> ### Author Response · Authors · 2021-08-10
> **Response to Reviewer 8yHH**
>
> Thank you for your time and suggestions! We will discuss all the suggested papers in our related work for the next version. Now, we will address your main concern as below:
>
> ### Q1: I could not identify a clear statement of the limitations of the paper anywhere in the paper. The word “limitation” does not appear in the paper.
>
> - Although the high-level idea of CO-PILOT is very general, i.e., a planner and an RL agent learn from each other’s feedback on an automatically generated curriculum of tasks, we only evaluate it on two types of tasks, i.e., maze navigation and robotic control. We will evaluate it on more diverse RL tasks and study if it can generally improve the training efficiency. Moreover, due to our limited computational resources, we have not tested it on more complicated tasks, e.g., the ones with dynamic environments or visual inputs such as games. It is still unclear whether it will still bring improvements to these tasks.
>
> ### Q2: The paper needs to more strongly articulate what is different about their approach, and what specific benefits are enabled.
>
> -  We summarize the differences and advantages of CO-PILOT compared to these methods:
> - Konidaris (“From Skills to Symbols”, 2018) studies the problem of planning a sequence of high-level actions using learned symbolic abstract representations. **In contrast, CO-PILOT does not need to pre-train such complex representations**, which is a challenging task. Instead, CO-PILOT takes feedback from an RL agent (trained together with the planner) to update the planning policy, automatically generating the subtasks. Moreover, **our learnable planning policy is more straightforward (given two endpoints, it produces a waypoint) but can be more potent than search-based planning in the abstract space**. By recursively applying our planning policy, we can achieve **multiple coarse-to-fine sequences of subtasks**, all connecting the starting and the goal states, while the planning in Konidaris et al. 's paper **only produces one high-level action sequence**. Furthermore, CO-PILOT is a mutual training scheme of planning and RL policies that uses one to improve the training of the other, while Konidaris et al. 's paper only focuses on planning.
> - PolCA (“Policy-contingent abstraction for robust robot control”, 2003) builds a hierarchical clustering of the state/subtask space and train an agent policy that determines a coarse-to-fine subtask (motion primitives) sequence for robot control on the hierarchy. In contrast, the hierarchy in CO-PILOT is a subgoal tree generated only for a single long-horizon task $(s,g)$: **it is not a hierarchical clustering/partition of all possible states/tasks**. Instead of being used as a hierarchical policy, the subgoal tree aims to provide a more efficient training curriculum for both a path-planning policy and an RL agent. The nodes (subgoals) in each level of the subgoal tree constitute **a trajectory of subgoals connecting $s$ and $g$. They are different from the nodes in PolCA, which are clusters of candidate subtasks**. That being said, the idea of CO-PILOT can be applied to the tasks of PolCA, i.e., our learned planning policy directly produces coarse-to-fine subtasks for a given task and thus can replace the costly hierarchical clustering and the tree search on it.
> - Michini (“Bayesian nonparametric inverse reinforcement learning”, 2012) decomposes the reward function learning of inverse RL into subproblems. It uses a generative model that can automatically partition the demonstrations into sets of smaller sub-demonstrations and then learns a reward function for each sub-demonstration. This can be costly for complicated environments or tasks since it needs to cover all possible states and value functions to guarantee traversal completeness. **While their method learns a posterior distribution of the value function, CO-PILOT does not aim to solve this inverse RL problem: it learns a planning policy to generate relatively coarse sequences of subgoals that guide the RL training**. Another difference is that our subgoal tree hierarchy is not a hierarchical partition of a demonstration space.
>
> ### Q3: What if the learner were replaced with a fixed set of motion primitives? Figure 4 might suggest that the algorithm is just learning a sampling distribution, and a comparison to “Learning Sampling Distributions for Robot Motion Planning” (Ichter et al, 2018) would be appropriate, especially if learning the controller inside the planning loop really does add something.
>
> - As shown in Fig. 1(a), **a subgoal tree is generated only for a single long-horizon task $(s,g)$, so it is not a hierarchical clustering/partition of all possible states/tasks and thus cannot be used for tree search like RRT or PolCA**. Moreover, the nodes in each level of a subgoal tree are not clusters of subtasks/states: they constitute a sequence of subgoals/waypoints connecting the starting state $s$ and the goal $g$ of the task $(s,g)$. So it cannot be used as a decision-tree policy as PolCA to determine which action the RL agent should take, though each subtask can be viewed as a motion primitive. The subgoal tree in CO-PILOT is applied to train a planning policy that aims to replace the expensive search-based planning/RL-policy and avoid the cost of hierarchical partition (of a large state/task space) by “learn to search/plan”. **Because CO-PILOT is not a search-based method, we do not need a sampling distribution**. One can think that our planning policy learns an implicit sampling distribution but does not use it for search.
> - That being said, CO-PILOT can be applied to solve the problems RRT and PolCA aim to solve and can potentially outperform them (it does outperform RRT* in the experiments). CO-PILOT can be directly applied to train an RL agent whose actions are from a fixed set of predefined motion primitives because it does not place any constraints on the action space. As an extension, the subgoals in CO-PILOT can even be replaced by the predefined motion primitives with only minor changes to the current algorithm, i.e., changing the inputs and output of the planning policy $\pi_p(g|g_i,g_j)$ to motion primitives. Compared to PolCA, CO-PILOT can decompose a given task into multiple subtask sequences of different granularity, which forms easy-to-hard curricula for training an RL agent or a planner. This enables the planner to generate new motion primitives and the RL agent to finish a long-horizon task by taking low-level actions with less guidance from the planner.
>
> ### Q4: CO-PILOT in high-dimensional state space.
>
> - **CO-PILOT can be applied to tasks in high-dimensional state spaces**, e.g., an agent with visual input. Since the training of the path-planning policy in CO-PILOT mainly relies on the time costs of the RL agent collected on different tasks, it can still work for high-dimensional state spaces with only minor changes. The only change needed is a pre-trained autoencoder producing low-dimensional representations of the high-dimensional states. This is a common strategy adopted by previous works [1,2] and can also be used to reduce the input and output dimensionality of the planning policy in CO-PILOT. In the paper, we have evaluated CO-PILOT on different dimensionalities for the state space. We will evaluate CO-PILOT in higher dimensional spaces once having sufficient computational resources for these tasks.
>
> ### Q5: The paper contains some statements and assumptions about planning that are not quite right.
>
> - We will rephrase those statements about planning and make them accurate.
>
> [1] https://proceedings.neurips.cc/paper/2020/file/c8d3a760ebab631565f8509d84b3b3f1-Paper.pdf
>
> [2] https://arxiv.org/pdf/1807.04742.pdf

---

> > ### Comment · Reviewer_8yHH · 2021-08-27
> > **Thank you for your response.**
> >
> > Thank you for your detailed response. Alas, I am left with more questions. I appreciate the comparison to the references I mentioned in my original review, but the authors really have not articulated what is the power of their approach. What can they do that the field as a whole cannot do? Also, I was puzzled by this sentence in their response: "One can think that our planning policy learns an implicit sampling distribution but does not use it for search." I don't know what one would do with the implicit sampling distribution except use it in search? I also would recommend the authors think about the limitations of the *approach*, not of the paper or their evaluation.
> >
> > I am still slightly predisposed to recommend this paper for acceptance, but I still see structural weakness in the argument for the technique.

---

> > > ### Author Response · Authors · 2021-08-27
> > > **Response to further questions: no search, no sampling distribution, no hierarchical/tree search, learning to plan, policy directly outputs subgoals**
> > >
> > > Thanks for your response! We are glad to hear that our response addressed some of your concerns. We understand that the reviewer is still **not clear about the essential difference of our method compared with previous hierarchical planning methods** and the reviewer is confused by one sentence in our response. Although we elaborated on our differences comparing to multiple methods in our response to Q2 and Q3, there are too many details and we believe it is necessary to provide a brief summary here.
> > >
> > > * The planning in our method is **"learning to plan" instead of search-based planning**. We train a policy to **directly outputs the planned trajectories of subgoals**. Instead, conventional planning does not learn such a policy and **has to search for subgoals in a huge space** among a great number of candidate states/actions, which is much more expensive and requires collecting more data. Our method avoids this expensive process.
> > >
> > > * Our method is **Not a planning method but a collaborative learning scheme between both planning and RL**: it makes both the path-planning and RL easier (due to denser and more informative feedbacks) by learning from each other on a curriculum of subtasks. This is an original and unique advantage of our method compared to previous works.
> > >
> > > * **No search happens in our method, so we do not use or learn any sampling distribution**. Since the reviewer mentioned in the original comment that "Figure 4 might suggest that the algorithm is just learning a sampling distribution", the sentence quoted in the reviewer's reply was trying to build a connection to the reviewer's original thought. But we would like to clarify that the original comment is inaccurate since we are not learning or using any sampling distribution.
> > >
> > > * **Our subgoal tree is Not the hierarchy in the previous hierarchical planning methods**, they are not even related. The previous hierarchy was defined to cover all possible subgoals/motion primitives and partition them into subsets: **it is usually very large, expensive to build, and aims to facilitate search-based planning**. In contrast, our subgoal tree is not used for searching subgoals but only for building curricula between a starting state $s$ and a final goal $g$: **it is much smaller, and its nodes only cover a few waypoints between $(s,g)$**. Given $(s,g)$, the first layer adds only one single waypoint between $s$ and $g$, and the second layer adds two more waypoints between $s$, the first-layer waypoint, and $g$, and so on. Again, these waypoints are directly generated by the planning policy rather than by search on any hierarchy. The layers in our subgoal tree are simply trajectories of subgoals connecting $s$ and $g$.
> > >
> > > We greatly appreciate your support and comments on our work! We will add more discussions about the limitations of our method. We hope that the above points can address your further concerns. Please feel free to let us know if you have any further questions and we are more than willing to respond and discuss.

---

> > > > ### Comment · Reviewer_8yHH · 2021-09-11
> > > > **Acknowledged**
> > > >
> > > > Thanks for the additional detailed reply. As I said before, I am still predisposed to recommend this paper for acceptance, and my concerns are around the argument. If the paper doesn't succeed, then I might suggest thinking through how better to tell your story, incorporating some of the points you included in your reply.

---

> > > ### Comment · Area_Chair_GETK · 2021-08-31
> > > **smoothing the way to understand what's going on in this paper**
> > >
> > > I was happy to see that reviewer 8yHH has written a very nice review, in terms of thoroughness and comprehensibility. Bravo! May I also note that I recently recruited another reviewer, qKpT, within the last few weeks and IMO this has turned out well also.

---

### Official Review · Reviewer_NJTw · 2021-07-17

**Rating:** 6
**Confidence:** 4

**Summary:**

This paper discusses how planning and goal-conditioned RL can be combined for long-horizon planning. The method proposes to use a recursive subgoal generating policy, which is applied repeatedly at k levels to create 2^k subgoals. A lower level goal-conditioned policy attempts to reach the subgoals. The objective for the higher-level policy depends on the lower-level policy by estimating the time it takes the lower-policy to go between a state and goal. The paper shows that the method outperforms planning methods (NEXT, RRT, SGTPG), RL methods (SAC, PPO) and hybrids (SoRB, HRL). Generally the paper is well-written and communicates effectively idea that each module can assist the other in these hybrid planning/GCRL methods. However, it is not fully clear to me yet from the experiments which design decisions were important, and the method is also only evaluated on relatively simple tasks.

**Limitations And Societal Impact:**

Limitations and societal impact are not really discussed. For limitations, it would be good to see a task where CO-PILOT fails or at least struggles to solve completely.

**Main Review:**

The paper proposes CO-PILOT, which introduces a recursive subgoal generating policy to make reaching far-away goals easier for a goal-conditioned policy. The claim is that the higher level policy assists the lower level policy by generating closer goals (standard HRL), but the lower level policy also assists the higher level policy by informing the cost function between states, which gets more accurate with a higher quality low-level policy as better data is collected.

Method: the writing is relatively clear. There is something cyclic going on that I didn’t understand in the top-down planning/algorithm 1. c is defined in terms of tau; tau is defined in terms of c_phi; c_phi is optimized … how? what is the objective? From algorithm 1 it seems like it depends on c. Is there a “grounded” cost that the whole system actually optimizes (does it come from alpha being 1 in eqn 12 so it is L2 distance)?

Experiments: The experiments show that CO-PILOT outperforms all other classes of methods: planning methods (NEXT, RRT, SGTPG), RL methods (SAC, PPO) and hybrids (SoRB, HRL). In section 4.4, additional visualization on a 2D maze shows the CO-PILOT learning process, splitting a goal into subgoals and improving the policy between subgoals. However, on this task, other methods are just as good; a more helpful visualization would be to understand why other methods fail on harder tasks while CO-PILOT does not. Perhaps a similar visualization can be constructed for something like Ant (still plotting on a 2D maze by accessing the underlying state information).

Generally, the paper proposes a large system of learning a cost function, a planner/subgoal generator, and a goal-conditioned policy. It is difficult to parse from the writing and results which component matters in the end, although the experiments are relatively thorough and may have the answer already. Some questions are: if SGTPT performs similarly to CO-PILOT, does this mean the goal-conditioned reward feedback to the planner does not matter (or what is the delta between the two)? The planner incorporates L2 reward, while the maze tasks are sparse reward; what if the L2 heuristic was used for SAC as the cost instead of sparse rewards?

Tasks: the paper evaluates on a few different domains, but one criticism is that the domains are all either maze tasks from state (2D, 3D, 5D, and ant maze locomotion) or simple (BipedalWalker). In contrast, there exists several works now that have shown the ability to do goal-conditioned planning in visual domains, including SoRB (cited and compared in this work, but not in visual domains), [2], [4].

The related work is relatively complete in terms of discussing goal-conditioned RL and planning. I did not quite understand how c_phi is optimized but it looks a lot like dynamical distance learning [1]. Also consider discussing [2] and [3] which explore subgoal generation, especially [2] which generates subgoals by divide-and-conquer exactly like this paper does.

[1] Dynamical distance learning for unsupervised and semi-supervised skill discovery. Hartikainen 2019.
[2] Long-Horizon Visual Planning with Goal-Conditioned Hierarchical Predictors. Pertsch 2020.
[3] Automatic Goal Generation for Reinforcement Learning Agents. Florensa 2018.
[4] ViNG: Learning Open-World Navigation with Visual Goals. Shah 2021.

**Time Spent Reviewing:**

4

---

> ### Author Response · Authors · 2021-08-10
> **Response to Reviewer NJTw**
>
> We appreciate for your time and suggestions! In this reply, we address your main concern as below:
>
> ### Q1: Limitations of CO-PILOT
>
> - Although the high-level idea of CO-PILOT is very general, i.e., a planner and an RL agent learn from each other’s feedback on an automatically generated curriculum of tasks, we only evaluate it on two types of tasks, i.e., maze navigation and robotic control. We will evaluate it on more diverse RL tasks and study if it can generally improve the training efficiency. Moreover, due to our limited computational resources, we have not tested it on more complicated tasks, e.g., the ones with dynamic environments or visual inputs such as video games. It is still unclear whether it will still bring improvements to these tasks.
>
> ### Q2: c is defined in terms of tau; tau is defined in terms of c_phi; c_phi is optimized … how? What is the objective?
>
> - The cost predictor $c_\phi(g,g’)$ is an MLP with three hidden layers. We train it via backpropagation with an objective minimizing the difference between its predicted costs for the subtasks $(g,g’)$ and the true time costs spent by the latest RL agent to finish those subtasks, which are collected in the bottom-up traversal of the sub-task tree (Alg. 2 L14).
>
> ### Q3: Is there a “grounded” cost that the whole system actually optimizes (does it come from alpha being 1 in eqn 12 so it is L2 distance)?
>
> - There is a “grounded” cost but it is not the L2 distance since it cannot take the obstacles into account and is not adaptive to the performance of the RL agent. Instead, our “grounded” cost is the time cost of the RL agent to finish a task, which precisely reflects the true cost and can be applied in general cases (i.e., task and environment agnostic, no prior knowledge is needed). In Eq. (12), we only use L2 distance at the very beginning of training to “warm start” the cost function when we have no time cost data collected from the RL agent. We quickly switch to the time cost by reducing $\alpha$.
>
> ### Q4: If SGTPT performs similarly to CO-PILOT, does this mean the goal-conditioned reward feedback to the planner does not matter (or what is the delta between the two)?
>
> - SGTPG only performs comparable to CO-PILOT on easier tasks like 2D planning in a simple environment (2D in Fig. 2(a)). As we increase the complexity of the RL agent and the environments, e.g., 3D/5D tasks (L605-610, Appendix B) in a complicated maze, SGTPG performs much poorer than CO-PILOT (3D/5D in Fig. 2(a) and Fig. 4(a)). The reason for this gap is that SGTPG uses Euclidean distance as its cost, which may work in simpler and low-dimensional space but cannot adapt to the RL agent and reflect the true cost. So the goal-conditioned feedback for the planner is important. The difference can also be seen in Fig. 4: from left to right (earlier to later training episodes), the Euclidean distance of subtasks becomes less uniform but the time cost to finish the subtasks drastically reduces and becomes more uniform.
>
> ### Q5: The planner incorporates L2 reward, while the maze tasks are sparse reward; what if the L2 heuristic was used for SAC as the cost instead of sparse rewards?
>
> - The L2 reward is not a proper reward. It is even harmful because it does not consider the obstacles: two points close in terms of L2 distance can have multiple obstacles between them, and the corresponding navigation is difficult. In contrast, the navigation between two distant points in terms of L2 distance can be much easier if there is no obstacle. In many RL tasks (e.g., the ones in this paper), a reward ignoring the obstacles can be catastrophic to the policy training.
> - Moreover, L2 reward is not adaptive to the RL agent and is irrelevant to its value function (the ideal choice for the dense reward (L149-151)), while the time cost can faithfully reflect the difficulty of a task to the RL agent. The only reason for us to use Euclidean distance in Eq. (12) at the very beginning of training is for “warm start” of the cost function when we have not collected any time cost from the RL agent.
> - In experiments, RRT* and SGTPG both use Euclidean distance as their cost function and perform poorer than CO-PILOT.
>
> ### Q6: Also consider discussing [2] and [3] which explore subgoal generation, especially [2] which generates subgoals by divide-and-conquer exactly like this paper does.
>
> - The subgoal generation in [2] also follows a top-down and coarse-to-fine manner and they still need to search for each subgoal in the tree from a set of sampled candidates. While we learn a planning policy to generate each subgoal from its parents directly and we do not need to build the search tree covering all possible subgoals. Another significant difference is that they study hierarchical planning, while we study the mutual learning between planning and RL on a subgoal hierarchy, which eases the training of both planning and RL policies.
> - [3] trains a GAN model to propose goals/tasks for goal-conditioned RL but each task is not decomposed into subtasks and each goal is not guaranteed to be a subgoal of other goals. It does not use path-planning to help the RL training and the GAN model can be difficult to train. In contrast, for each task, CO-PILOT builds two curricula along a structured subgoal tree: one trains the RL agent to accomplish the task with more guidance (from the planner) at first and then switches to less guidance, and the other trains the planning policy from easier planning tasks (fewer waypoints) to harder ones (more waypoints). Along the curricula, they learn from each other’s feedback. This mutual training overcomes the bottlenecks of training RL or planning policy independently (e.g., dense rewards for both).
>
> [1] Dynamical distance learning for unsupervised and semi-supervised skill discovery. Hartikainen 2019.
>
> [2] Long-Horizon Visual Planning with Goal-Conditioned Hierarchical Predictors. Pertsch 2020.
>
> [3] Automatic Goal Generation for Reinforcement Learning Agents. Florensa 2018.

---

> > ### Comment · Reviewer_NJTw · 2021-09-11
> > **Response to Rebuttal**
> >
> > Thanks for the thorough response. My doubts were clarified; the limitations of the evaluation domains still persist though and I will keep my score the same.

---

> ### Author Response · Authors · 2021-09-01
> **[Last day reminder] We have addressed all your concerns with detailed answers. Would you mind checking them and reconsidering your rating?**
>
> Dear Reviewer NJTw：
>
> Thanks a lot for your original comments! We have addressed every concern from you with detailed answers in the response below posted three weeks ago. Would you mind checking them and confirming if there is any concern left? We humbly expect that you can reconsider your ratings given all these answers. Your support is very important to our work and we greatly appreciate that!
>
> Best Regards,
>
> Authors

---

### Official Review · Reviewer_qKpT · 2021-07-25

**Rating:** 6
**Confidence:** 4

**Summary:**

**Summary:** This paper proposes CO-PILOT that can improve performance in goal-conditioned RL settings. The main contribution of CO-PILOT is the collaborative learning between the planning and RL agents: the planning agent generates the sub-task tree providing dense feedback for the RL agent to solve a task, while the RL agent collects data and refines the cost function for the planning agent. Appealing benefits of CO-PILOT include 1) the auto-generated curriculum that helps training both the planning and RL agents, and 2) no requirements of a pre-trained policy, a heuristic for the cost function, and an accurate environment model. Empirical results show that CO-PILOT outperforms RL, planning, and mixed baselines in various domains.

**Limitations And Societal Impact:**

This paper does not discuss any potential negative societal impacts.

**Main Review:**

**Reasons for Score:** Overall, I vote for a score of 6. While CO-PILOT provides an interesting insight (i.e., the mutual boosting between RL and planning) and shows positive evaluation results, there are several concerns and questions that I would like to clarify (please refer to Concerns and Questions below). After reading the authors' responses to my concerns, I am open to raising my score.

**Strengths:**
1. The paper is well written and motivates the main insight very well.
2. The approach is principled. In particular, the paper introduces the interesting idea of auto-generated curriculums for the planning agent (top-down) and RL agent (bottom-up) to stable learn these policies.

**Concerns and Questions:**
1. Novelty is limited because CO-PILOT shares many similarities to the sub-goal tree method [24], except for the additional learning of the RL agent.
2. While the result in Figure 2a show CO-PILOT outperforms the sub-goal tree baseline (SGTPG), it is unclear from the result discussion in Section 4.3 how CO-PILOT achieved this result. It will be helpful to include a discussion on what algorithmic difference resulted in better performance.
3. Did the simultaneous learning of planning and RL agents result in a non-stationarity issue? For example, in [37], the simultaneous learning of both low and high-level policies created a non-stationary problem and they addressed the issue in an off-policy learning setting.
4. Many planning methods have important properties, such as asymptotic optimality, probabilistic completeness, and monotone convergence (Karaman and Frazzoli, 2011). Does the planning agent in CO-PILOT also maintain these properties?
5. One main benefit of CO-PILOT is the curriculum learning for the planning/RL agents. This benefit is also presented in the hindsight experience replay (HER) approach [3], where the goals used for replay naturally shift from simple to difficult ones. While it is not needed to compare against HER during the rebuttal period, it would be helpful to include the comparison against HER in the future.

**Additional Feedback:**
1. Similar to having $\pi$ as a superscript in the Q-function $Q^{\pi}$ (denoting the agent follows $\pi$ in the future states), it will be helpful to denote the value function as $V^{\pi}$ instead of $V$.
2. It will be helpful to add variances in some results: Figures 2a, 2c, 4a, 4b, 4c.
3. It will be helpful to clarify whether the results (e.g., Figure 2a) are from training or testing pairs.
4. Figure 1 is not discussed in the introduction, so it would be better to place the figure where it is referenced (Section 3.3). Additionally, Figure 1b is not referenced in the text.
5. The current pdf format does not support search (i.e., ctrl + f) (tried on Linux and Mac). It was difficult to find a specific keyword in the paper, so I hope this issue is fixed. This might be not an issue with other computers.

**Reference:**
Sertac Karaman and Emilio Frazzoli. Sampling-based Algorithms for Optimal Motion Planning. International Journal of Robotics Research (IJRR), 2011.

**Post-Rebuttal:** Please refer to the review below.

**Time Spent Reviewing:**

8

---

> ### Author Response · Authors · 2021-08-10
> **Response to Reviewer qKpT**
>
> We appreciate your time and suggestions! As you suggested, we report the results of new experiments of the comparison to HER. In this reply, we address your concerns except the experimental comparison:
>
> ### Q1: Novelty is limited because CO-PILOT shares many similarities to the sub-goal tree method
>
> - **Our main novelty is not the subgoal tree** (we only use it to define our planning policy). **CO-PILOT is different from sub-goal tree method in several fundamental aspects**, e.g., the targeted problem, the main contribution, the training procedure, etc:
> - **They target different problems**: sub-goal tree method is a planning method, while CO-PILOT is mainly an RL method. The path-planning in CO-PILOT aims at improving and accelerating the RL training.
> - **Our main novelty is the collaborative/mutual learning between the path planner and the RL agent**, where each can provide dense feedback to train the other and thus overcome the bottlenecks of both planning-only and RL-only approaches. **The generated sub-goal tree in our method is used to build two easy-to-hard curricula for the two models** (bottom-up for RL and top-down for planner), respectively. **These were not studied in the sub-goal tree method**: they do not adopt any curriculum (they train different policies for different levels of the tree) or mutual training scheme.
> - **The training procedure for the sub-goal tree policy is different**: sub-goal tree method independently trains an SGT policy for every level using Euclidean distance as the cost, while CO-PILOT trains one single planning policy for all levels using the feedback from the RL agent as the cost.
> - In all the experiments, **CO-PILOT outperforms the sub-goal tree method by a large margin**.
>
> ### Q2: CO-PILOT outperforms the sub-goal tree baseline (SGTPG), it is unclear from the result discussion in Section 4.3 how CO-PILOT achieved this result
>
> There are several reasons for the superior performance of CO-PILOT over SGTPG:
> - **CO-PILOT trains the planning policy using the feedback from an RL agent and follows an easy-to-hard curriculum** (via top-down construction in Alg. 1). The former provides more accurate training signals, while the latter makes the training more efficient. In contrast, SGTPG uses Euclidean distance as the cost and trains an independent planning policy for each level (no curriculum across levels), which leads to a poorer planning policy.
> - **CO-PILOT conducts mutual training between a path-planner and an RL agent**. The RL agent is trained using the dense reward from the planner by following an easier-to-harder curriculum on the subgoal tree. The planner and RL agent mutually boost each other’s performance and they together result in the superior performance of CO-PILOT over SGTPG. In contrast, SGTPG only learns planning policies and cannot benefit from the RL agent’s training or the mutual learning.
> - **In SGTPG, they only build the subgoal tree once for each sampled (starting point, goal) pair in training, while CO-PILOT repeatedly refine the subgoal tree** of each state-goal pair (Alg. 3 L6-9) for multiple episodes to train a higher-quality and more adaptive planning policy.
>
> ### Q3: Did the simultaneous learning of planning and RL agents result in a non-stationarity issue?
>
> - **We do not have the non-stationary issue of hierarchical RL** because our path-planning policy (analogous to the high-level policy) and its generated subgoal tree (analogous to the high-level actions) **always adapt to the latest RL policy** (analogous to the low-level policy) during training. As shown in Alg. 3, in each episode, we optimize the planning policy until convergence using the cost data collected by the latest RL agent policy, and then optimize the RL policy until convergence using the dense reward defined by the latest planning policy. We do not need to relabel the stale high-level actions as [37] because we remove the old high-level actions generated for previous episode’s low-level policy (Alg. 2 L3). **One main difference of our method compared to previous HRL is the tree-structured curricula, which provides more informative experiences to train the two policies**.
>
> ### Q4: Does the planning agent in CO-PILOT also maintain properties as many planning algorithms? (probabilistic completeness, and monotone convergence (Karaman and Frazzoli, 2011))
>
> - **Instead of using dynamic programming for path-planning, we study “meta-planning” or “learn to plan”** to solve the same problem, i.e., training a planning policy to directly generate the shortest path between any valid state-goal pair. **It is not straightforward to analyze whether the meta-trained policy has the properties of online optimization**. Compared to the dynamic programming method, our inference of the path can be more efficient and adaptive to the RL agent, which perfectly fits our needs of constructing the curricula. It is an open problem to study the optimization properties of meta planning and we will continue to work on its theory in the future.
>
> ### Q5: Whether the results (e.g. Figure 2a) are from training or testing pairs?
>
> - Fig 2.(a) reports the performance of CO-PILOT and baselines **on the test pairs only**, which are different from the training set (as we introduced in L300).
>
> ### Q6: Comparison to HER in the Maze environment
>
> - Both HER and CO-PILOT train a goal-conditioned RL policy on modified/new tasks with dense rewards. Their main difference is that **HER relabels the failed tasks tried in history while CO-PILOT trains a sub-goal planning policy to generate a curriculum of new subtasks**. Compared to HER that cannot fully control the utility of the relabeled tasks, CO-PILOT learns to propose subtasks adaptive to the RL agent’s training progress and automatically organizes these subtasks in an easy-to-hard curriculum, which provides more informative and effective guidance to RL. This difference usually leads to a higher success rate and better training efficiency of CO-PILOT, as shown in the experiments below.
> -**We present a new empirical comparison of HER with CO-PILOT on the Maze environment tasks**. We chose a highly rated implementation of HER on Github and evaluated “DQN with HER” and “DDPG with HER”. For a fair comparison, HER is trained on 300 $(s_0,g)$ pairs and tested on 100 $(s_0,g)$ pairs (the same as CO-PILOT, Sec. 4.1). We apply the training setting used in our paper for HER.
>
> - **Table 1** The success rate averaged over 6 test epochs for 100 test tasks (mean$\pm$std) in Maze (2D/3D/5D defined in Sec. B of the Appendix).
> |Method|Maze (2D)|Maze (3D)|Maze (5D)|
> |:--|:--|:--|:--|
> | CO-PILOT | 94.02$\pm$3.37 | 87.34$\pm$5.25 | 78.26$\pm$8.24 |
> | DQN with HER | 89.12$\pm$5.34 | 82.58$\pm$6.14 | 78.81$\pm$10.62 |
> | DDPG with HER | 80.85 $\pm$ 7.67 | 70.02$\pm$10.37 | 66.43 $\pm$ 11.25|
>
> - In the table above, **CO-PILOT outperforms HER on most tasks except on Maze(5D)**, where DQN with HER has a slightly higher mean success rate and CO-PILOT has a smaller std (more stable performance). Due to the limited time for rebuttal, we have not evaluated HER in more environments (such as Mujoco-Ant and BipedalWalker in the paper) but we will add more comparison results later.
>
> [1] https://jeffe.cs.illinois.edu/teaching/algorithms/book/09-apsp.pdf
> [2] https://www.sciencedirect.com/book/9780124170490/introduction-to-mobile-robot-control (section 11.3)

---

> > ### Comment · Reviewer_qKpT · 2021-08-26
> > **Review for CO-PILOT (After Author Responses)**
> >
> > Thank you for your response. The authors have clarified my questions about the paper. I also appreciate the authors for conducting the additional comparison against HER.
> >
> > After carefully reading other reviews and authors' responses, I agree that the main idea, the mutual learning between the path planner and the RL agent, is interesting. However, I also agree with potential concerns, including 1) relatively simple experimental evaluations w.r.t. recent papers that have more MuJoCo domains and/or visual domains and 2) insufficient analysis of CO-PILOT (e.g., which component (auto-generated curriculum, the adaptive cost function, sub-goal tree, etc) was crucial in improving performance). Hence, I generally recommend this paper for acceptance, but I don't argue strongly for it and I keep my score.

---

### Decision · Program_Chairs · 2021-09-28

**Decision:**

Accept (Poster)

**Comment:**

The reviewers' appreciate the interesting ideas in this submission and the generally clear presentation, and the authors detailed responses to the reviewers' concerns. The authors are encouraged to revise their camera ready to address the feedback from the reviewers.

**Consistency Experiment:**

NeurIPS has a long history of experimentation. In 2014, NeurIPS ran an experiment in which 10% of submissions were reviewed by two independent committees to quantify the randomness in the review process. This year, we repeated a variant of this experiment to see how the quality of the review process has changed over time.  This paper was part of the experiment and was therefore assigned to two committees (consisting of reviewers, an Area Chair, and a Senior Area Chair) that reached independent decisions.  If both committees made the same recommendation, this recommendation was followed. If a single committee recommended acceptance, the paper was accepted (with the exception of a few cases in which the other committee identified what we considered a fatal flaw, e.g., an error in a key result).

This copy’s committee reached the following decision: **Accept (Poster)**

The other committee assigned to the paper recommended **Reject**.  You can find the other set of reviews, along with any follow up discussion with the authors here:
https://openreview.net/forum?id=uz_2t6VZby